# The Association between Stressful Life Events and Emotional and Behavioral Problems in Children 0–7 Years Old: The CIKEO Study

**DOI:** 10.3390/ijerph19031650

**Published:** 2022-01-31

**Authors:** Yuan Fang, Hein Raat, Dafna A. Windhorst, Irene N. Fierloos, Harrie Jonkman, Clemens M. H. Hosman, Matty R. Crone, Wilma Jansen, Amy van Grieken

**Affiliations:** 1Department of Public Health, Erasmus University Medical Centre, 3000 CA Rotterdam, The Netherlands; y.fang@erasmusmc.nl (Y.F.); dafna.windhorst@tno.nl (D.A.W.); i.fierloos@erasmusmc.nl (I.N.F.); w.jansen.1@erasmusmc.nl (W.J.); a.vangrieken@erasmusmc.nl (A.v.G.); 2Department of Cognitive Neuroscience, Donders Institute for Brain, Cognition and Behavior, Radboud University Medical Center, 6525 GD Nijmegen, The Netherlands; 3TNO Child Health, 2316 ZL Leiden, The Netherlands; 4Verwey-Jonker Institute, 3512 HG Utrecht, The Netherlands; hjonkman@verwey-jonker.nl; 5Department of Health Promotion, Maastricht University, 6229 HA Maastricht, The Netherlands; Hosman@psych.ru.nl; 6Department of Clinical Psychology, Radboud University, 6525 XZ Nijmegen, The Netherlands; 7Hosman Prevention and Innovation Consultancy, 6562 DW Berg en Dal, The Netherlands; 8Department of Public Health and Primary Care, Leiden University Medical Center, 2333 ZA Leiden, The Netherlands; m.r.crone@lumc.nl; 9Department of Social Development, Municipality of Rotterdam, 3000 LP Rotterdam, The Netherlands

**Keywords:** life events, emotional and behavioral problems, internalizing behaviors, externalizing behaviors, children

## Abstract

Background: Stressful life events (SLEs) are recognized risk factors for emotional and behavioral problems, but the association is understudied among young children. Our aim was to examine the association between exposure to SLEs and emotional and behavioral problems in young children up to 7 years old. Methods: We analyzed baseline data from 959 children (mean age = 3.3 years; SD = 1.9; 47.5% girls) in the CIKEO study, a community-based longitudinal study in the Netherlands. Linear regression was used to assess the associations between the total as well as the individual exposure to SLEs experienced in the past 12 months, and emotional and behavioral problems assessed by CBCL 1.5-5. Interactions of SLEs and child age, sex, ethnic background, and socioeconomic status were explored. Results: Higher total exposure to SLEs, as indicated by the number of SLEs, was significantly associated with higher CBCL total, internalizing and externalizing problem scores (*p* for trend < 0.05). The results did not differ by child age, sex, ethnic background, or family SES. Six out of the 12 SLEs explored were independently associated with greater CBCL total/externalizing/internalizing scores (*p* < 0.05). Conclusions: Exposure to SLEs is associated with higher levels of emotional and behavioral problems in young children, and the impact of SLEs may vary depending on the types of events. Stressful life events might be a useful target for interventions to improve emotional and behavioral well-being among young children.

## 1. Introduction

Emotional and behavioral problems, such as disruptive behavior, depression, and anxiety, characterized as either internalizing or externalizing problems, are common in childhood and affect up to 20% of children aged 1 to 7 years old [1]. These problems have been shown to be associated with adverse outcomes, such as poor educational attainment and difficulties in social adjustments in children [2,3,4,5,6], and can persist into adulthood [4]. Insight into the risk factors related to the development of emotional and behavioral problems may contribute to the development of effective support and timely interventions.

Stressful life events (SLEs) are a series of events that fall outside an individual’s normative life experiences [2]. The experience of SLEs in childhood has been linked to increased risk of emotional and behavioral problems later in life [3,7,8,9,10,11,12]. Examples of SLEs are financial problems, family and personal conflicts, and stressors related to health. The reported prevalence of SLEs in children is high and varies between studies [3,7,8,9,13]. In Europe, Vanaelst, et al. reported that about 40.3% of children aged 4–10 years have experienced at least one SLE [13]. The association between SLEs and emotional and behavioral problems can be studied using a ‘cumulative risk’ approach in which the number of stressful life events reported is summed up; the associations between this total score and an individual’s emotional and behavioral problems can be assessed [7,8,9]. Studies using this approach have shown a strong link between the total number of SLEs experienced and adverse outcomes in child and adolescent psychopathology, including a higher risk for externalizing and internalizing problems [3,14,15,16,17,18,19]. A limitation of this approach is the assumption that all life events have an equal impact on health and well-being; the impact may vary per event [20,21]. To address this, some studies used a weighted SLE score by taking the information about the severity or potential impact of the SLEs into consideration [17].

The associations between SLEs and emotional and behavioral problems can also be studied by evaluating the association of each specific SLE with emotional and behavioral problems separately. The results of prior studies have suggested that also some specific SLEs might be sufficient to trigger the emergence of psychological problems [11,21]. For instance, in a German study of children aged 5.0–6.9 years old, Furniss et al. observed that children who experience the life event “move of best friends”, were more likely to display internalizing problems. Meanwhile, externalizing problems were more prominent among children whose “parent lost a job” [3].

Despite the numerous studies in adults, adolescents, and older children, the role of SLEs in the development of emotional and behavioral problems in young children is still understudied [3]. Nevertheless, young children might be more vulnerable to SLEs, and early life exposure to SLEs might introduce more profound and long-lasting adverse effects on health [21,22,23]. In addition, the associations between SLEs and emotional and behavioral problems could also be different in subgroups of children according to sociodemographic characteristics (sex, ethnic background, and the socioeconomic status (SES) of the child). Yet, evidence on the moderating effects of these sociodemographic characteristics on the association have been inconclusive [12].

Therefore, using a community sample in the Netherlands, we aimed to examine the association between SLEs and emotional and behavioral problems in young children up to 7 years old. Specifically, we evaluated the association between (1) individual SLEs (yes/no) and, (2) multiple SLEs (i.e., the number of SLEs) with emotional and behavioral problems in children up to 7 years old. In addition, we explored interactions by child age, sex, ethnic background, and family SES.

We hypothesized that the association with emotional and behavioral problems is different for each specific SLE. In terms of the total number SLEs to which a child was exposed, we hypothesize that exposure to a higher total number of SLEs was associated with more emotional and behavioral problems. In line with previous studies [12,24], a higher risk for emotional and behavior problems when experiencing multiple SLEs was expected among girls, children from a minority ethnic background, and children from low-SES families.

## 2. Materials and Methods

### 2.1. Study Design and Data Collection

In this study, we applied a cross-sectional design using the baseline data from the CIKEO study. The CIKEO (Consortium Integration Knowledge promotion Effectiveness Of parenting interventions in the Netherlands) study is a community-based study with a baseline and a follow-up measurement [25]. The CIKEO study investigated the use of (elements of) parenting support and the associations between parenting support and outcomes regarding parenting, family functioning, and child development. Details of the study design including the recruitment procedure have been described elsewhere [25]. In brief, parents/caregivers with at least one child that is up to 7 years old were invited to participate in the study between October 2017 and December 2019. Participants were recruited in two parts. Participants in Part A were recruited by two regional preventive youth healthcare organizations in the regions of Rotterdam (CJG Rijnmond) and Dordrecht (RIVAS Zorggroep). Participants in Part B were recruited by providers of parenting programs or through advertisements on websites about parenting. All invited parents/caregivers received project information, an informed consent form, and a baseline questionnaire. Those parents/caregivers who spent the most time with the child were asked to complete the questionnaire. Parents who provided written, informed consent and completed a baseline questionnaire were included. A follow-up measurement was conducted after 12 months of enrollment using questionnaires.

### 2.2. Ethics

The Medical Ethics Committee of the Erasmus Medical Center, Rotterdam decided that the rules laid down in the Dutch Medical Research Involving Human Subjects Act (in Dutch: Wet Medisch-wetenschappelijk Onderzoek met mensen) did not apply to this study, that there were no objections to the execution of this study (proposal number MEC-2017-432), and approved the submission of the results of the study to scientific journals (Letter NL/sl/321518; 24/07/2017). The CIKEO cohort study was registered in the Netherlands Trial Registry (number: NL7342).

### 2.3. Study Population

In total, 1118 parents provided informed consent at baseline. For the study reported here, data from children was excluded when the questionnaires were not filled out for the same child (*n* = 18), and when there was missing data on the variables assessing stressful life events (*n* = 68) or the outcome measures (*n* = 73). Hence, 959 children were included in the analyses of this study (Appendix A).

### 2.4. Measures

#### 2.4.1. Child Emotional and Behavioral Problems

Emotional and behavioral problems were assessed using the 100-item Child Behavior Checklist for ages 1.5 to 5 (CBCL) [26,27]. As the majority (73.9%) of the children included in the current study fell in the range of 1.5–6 years old at the time of CBCL assessment (21.5% were younger than 1.5 and 3.6% of the children were older than six years old), we used the CBCL 1.5–5 version for all children to enhance comparability [28]. Parents rated the occurrence of their child’s behavior within the past 2 months on a three-point scale with 0 (not true) and 2 (very true or often true). The CBCL includes a total problem score (CBCL-T), and two broadband scales: internalizing and externalizing. The internalizing scale (CBCL–I) assesses behaviors such as withdrawal, anxiety, and depression; the externalizing scale (CBCL–E) assesses behaviors such as attention problems and aggressive behavior. A weighted sum score allowing for 20% of missing data was calculated for the total scale and the two broadband scales by summing the items belonging to each (sub)scale. Higher scores indicate higher levels of problems. To illustrate the clinical significance, we also calculated the percentage of children with a score above the borderline and clinical range. The cutoff points were based on the 83rd percentile based on a Dutch norm group (Table 1) [29].

#### 2.4.2. Stressful Life Events

At baseline, parents were asked if the following 12 life events had occurred in the family within the last 12 months. These life events were based on existing literature on this topic [30]: (1) moving to another address; (2) a friend of the child moving to another address; (3) tension at the parents’ work that has been felt at home; (4) financial problems; (5) conflicts with neighbors, friends, acquaintances, or family; (6) fire or burglary; (7) problems with the physical health of people in close proximity; (8) problems with the psychological health of people in close proximity; (9) death of someone in close proximity; (10) problems in the marriage relations; (11) divorce; (12) unemployment. The correlation between these life events was considered low (Spearman’ rho 0.39). Parents responded with yes or no for each item. If they responded with yes, they were asked to rate the severity of the event caused in the family (1 = not at all stressful, 2 = somewhat stressful, and 3 = highly stressful). When the parents gave the response ‘no’ to the question of whether a specific SLE occurred, a score of ‘zero’ was allocated for the ‘severity’ of that SLE.

For the current analyses, three types of variables regarding the presence and severity of SLEs were created. First, we made 12 variables to reflect the presence (no/yes) of each of the 12 SLEs; these variables were used to assess the associations between the exposure to a certain individual life event and the outcome measures in this study. Second, for the main analyses, the variable ‘number of SLEs’ was calculated by counting the number of SLEs that were reported to be present by the parent (possible range: 0–12). Next, based on previous literature [3], this score was recoded into the following categories: none (0 SLEs), 1 SLE, 2 SLEs, and 3–12 SLEs. Third, for the sensitivity analyses, the variable “overall severity experienced” was calculated by adding up the reported severity for each of the 12 SLEs (0 = not exposed, 1 = not at all stressful, 2 = somewhat stressful, and 3 = highly stressful); the potential range of the sum score is 0–36. The “overall severity experienced” was recoded into the following categories based on tertiles of the sample: no experience of SLE (score 0), low-severity SLE experience (score 1–2), medium-severity SLE experience (score 3–5), and high-severity SLE experience (score 6–36).

Covariates data on child age, sex, ethnic background, parental age, sex, family composition (single parent/ two-parent), and family SES indicators (i.e., maternal educational level) were obtained for each child through parent-reported questionnaires. A child’s ethnic background was defined as Dutch if both parents of the child were born in the Netherlands; otherwise, the child was considered non-Dutch [31]. Parental educational level was classified into “low” (no education, primary school/primary education/preparatory secondary vocational education), “medium” (senior general secondary education, pre-university education, and secondary vocational education), and “high” (higher vocational education/university). Net monthly household income was classified into three categories based on the income tertiles of the sample: low (<€3200), medium (€3200–4400), and high (>€4400).

### 2.5. Statistical Analysis

First, descriptive statistics were used to present the characteristics of the participants in the study. Second, linear regression was used to investigate the association between ‘number of SLEs’ (none, low, medium, high) and the CBCL 1.5–5 total, externalizing, and internalizing problems’ scores. The models were adjusted for child age, sex, ethnic background, respondents’ age, educational level, family composition, and household income. In addition, the recruitment method (Part A/Part B) was included as a potential confounder in all models. Testing for linear trends across the four groups of ‘number of SLEs’ (none, low, medium, and high) was performed by entering a single ordinal term. Subsequently, we tested sociodemographics (i.e., child age, sex, ethnic background, parental educational level) by SLEs interactions. Stratified analysis was performed if the interaction term was significant (*p* < 0.10 [32]). Next, a series of multiple regressions were performed to examine the association of each individual SLEs (no/yes) with CBCL-T/I/E scores. To examine the independent effect of individual SLEs, we additionally adjusted for the effect of other SLEs (i.e., the number of other SLEs). With regard to the exploratory approach of these analyses, no alpha adjustment was done.

### 2.6. Sensitivity Analysis

A sensitivity analysis using “overall severity experienced” was performed to investigate the association between multiple SLEs exposure and CBCL-T/I/E scores in children. The results were comparable to the main analysis (see Appendix A).

Missing values varied between 0.1% for parental age and 5.6% for household income (Table 1). Multiple imputation was applied to all variables included in this study using the R package ‘mice’. Five imputed datasets were generated for pooled estimates. All statistical analyses were performed using R version 3.6.6. Due to the skewed distributions of the CBCL scores, bootstrapped confidence intervals (95% CI) with 1000 iterations were computed for the coefficients [33].

## 3. Results

### 3.1. Nonresponse Analysis

The characteristics of the participants included in the final sample (*n* = 959) and those excluded due to missing data (*n* = 141) were compared. Children excluded often had parents who were single, unemployed, less educated, had lower income, and a non-Dutch ethnic background (*p* < 0.05). No other significant differences were found between these two groups.

### 3.2. Sample Characteristics

Child and family characteristics are summarized in Table 1. The sample consists of 959 children—499 boys (52%) and 456 girls (48%) with a mean age of 3.3 years (SD = 1.9, range: 0–7 years). The majority of the sample were Dutch (88.1%) and were from parents who were highly educated (56.5%), employed (82.8%), living with a partner (94.3%), and had a medium (41.0%) to high income (27.1%). There was no difference in exposure to SLEs with regard to child ages, sex, ethnic background household income, and parental educational level.

The mean scores of CBCL-T, CBCL-E, and CBCL-I were 20.3 ± 16.7, 8.8 ± 7.1, and 4.8 ± 5.3, respectively. Out of the 959 children, thirty-eight (4.0%) children scored in the “clinically significant” range for total emotional and behavioral problems. Seventy-seven (8.0%) scored in the “borderline” range for total emotional and behavioral problems. Children exposed to more SLEs reported higher CBCL-T, CBCL–I, and CBCL–E scores (*p* < 0.05) (Table 1).

### 3.3. Prevalence and Co-Occurrence of Stressful Life Events

The majority (72.8%, *n* = 699) of the family experienced one or more SLEs, and 27.1% (*n* = 260) had experienced three or more SLEs in the past 12 months. Parents reported that the family experienced, on average, 1.8 SLEs (SD = 1.7; range = 0–12).

The frequency and the severity rating for each life event are presented in Table 2. The most prevalent reported SLEs included tension at the parents’ work that has been felt at home (36.8%, n = 353), problems with the physical health of people in close proximity (36.2%, n = 347), death of someone in close proximity (22.3%, *n* = 214), and problems with the psychological health of people in close proximity (19.7%, *n* = 189). The least commonly experienced SLEs were parental divorce (2.7%, *n* = 26) and being a victim of fire or burglary (2.5%, *n* = 24).

### 3.4. Individual SLE and Emotional and Behavioral Problems

The associations between individual stressful life events and emotional and behavioral problems in children are presented in Table 3. Six out of the 12 SLEs were consistently associated with higher CBCL-T/I/E scores. After further adjustments for the effects of other SLEs, most of these associations attenuated or disappeared. Individual life events including “tension at the parents’ work that has been felt at home”, “conflicts with neighbors, friends, acquaintances, or family” remained consistently associated with a higher CBCL total [β (95% CI) = 4.52 (2.22, 6.83); 4.61 (1.05, 8.18)], externalizing [β (95% CI) = 2.05 (1.05, 3.06); 1.70 (0.14, 3.25)], and internalizing [β (95% CI) = 1.04 (0.34, 1.75); 1.42 (0.34, 2.50)] scale scores. Children who had experienced parental divorce also had increased CBCL-Total scores [β (95% CI) = 8.23 (0.33, 16.12)]. Three life events, namely “financial problems”, “problems in the marriage relations” and “divorce” were associated with higher CBCL-internalizing scores, with β and (95% CI) being 1.29 (0.02, 2.56), 1.33 (0.08, 2.57), and 4.65 (2.26, 7.03), respectively.

### 3.5. Multiple SLEs and Emotional and Behavioral Problems

Results of the regression analyses for each of the three outcomes (CBCL-T, CBCL-I, and CBCL-E) are summarized in Table 4. The adjusted model for CBCL-T showed that compared to children who experienced no SLEs, children exposed to low, medium, and high numbers of SLEs had higher CBCL-T scores, with beta and 95% CI being 3.86 (0.96, 6.70), 4.29 (1.45, 7.29), and 7.10 (4.36, 10.14), respectively. Moreover, the association was dose–response, with higher exposure being associated with more emotional and behavioral problems (*p* for trend < 0.001).

The model for CBCL-E yielded similar results. Compared with children who experienced no SLEs, children in the subgroups of low, medium, and high numbers of SLEs all reported higher CBCL-E scores, with β and 95% CI being 1.95 (0.67, 3.20), 1.57 (0.33, 2.90), and 3.12 (1.86, 4.50), respectively.

Compared to children who experienced no SLEs, children exposed to medium and high numbers of SLEs also reported higher CBCL-I scores, with β and 95% CI being 1.30 (0.43, 2.24) and 1.59 (0.75, 2.46), respectively. No significant association was found between exposure to low numbers of SLEs and CBCL-I score [β, 95% CI: 0.73 (−0.11,1.57)].

### 3.6. Interaction by Child Age, Sex, Ethnic Background, and SES

No interactions by child age, sex, ethnic background, and SES were found in the association between the number of SLEs and CBCL scores (*p* > 0.10 [32]). This suggests that the association between stressful life events and emotional and behavioral problems was not moderated by these sociodemographic characteristics.

## 4. Discussion

Using a community-based study among 959 young children up to 7 years old, we investigated the association between stressful life events and emotional and behavioral problems. Approximately 70% of the families reported one or more life events. Our data showed a ‘dose–response’ association, i.e., children exposed to a higher total number of SLEs generally had higher parent-reported total, externalizing, and internalizing problem scores. The impact of SLEs may differ by the event. The SLEs “work–family spillover” and “conflicts with neighbors, friends, acquaintances, or family” were associated with relatively more total, externalizing and internalizing problems. The SLEs “financial problems”, “problems in the marriage” and “divorce” were associated with relatively more internalizing problems.

### 4.1. Individual SLEs and Emotional and Behavioral Problems

Our study showed that exposure to certain individual SLEs might also increase the risk of emotional and behavioral problems in young children. Specifically, children exposed to events including “tension at parents work that has been felt at home”, “conflicts with neighbors, friends, acquaintances, or family”, “problems in the marriage relations”, “divorce”, and “financial problems” might have an increased risk for emotional and behavioral problems. Of note, we observed that while some events (e.g., “tension at parents’ work that has been felt at home”) are common in our sample; other events that are less common might be more influential to children. For instance, the CBCL total scores increased on average by 8.23 points in our sample when parents divorced. In terms of the potential effects on emotional and behavioral health in children, preventive interventions and policies should adequately address the needs of children who experienced/are experiencing these specific life events, especially those less common but influential events. However, due to the relatively small numbers in the analyses and explorative nature, our findings on individual life events have to be interpreted with care. We recommend future studies to verify our findings.

### 4.2. Multiple SLEs and Emotional and Behavioral Problems

The observed findings indicating a stronger association between higher numbers of SLEs and higher CBCL total, externalizing, and internalizing problem scores, are in line with previous studies [3,14,15,16,17,18,19]. Furthermore, this dose–response association was also observed in our additional analysis using the borderline and clinical cut-off points of the CBCL scores (data not shown). However, due to the small sample size in these subgroups, only children that experienced three or more SLEs had increased borderline and clinical problems. Several mechanisms or pathways underlying these associations have been suggested. Some of these pathways follow a biological explanation. For example, alterations in affective regulation, neurohormonal systems (e.g., changes in the output or tissue effects of hormones such as cortisol, testosterone, and estrogen), and activation of endocrine axes, including the sympathetic–adrenal–medullary (SAM) system and the hypothalamic–pituitary adrenal (HPA) axis [21,34]. Other pathways suggested are through the adverse effects of SLEs on parents, for instance, SLEs impacting parental psychopathology, parenting practices, and dynamics in the family [35,36].

### 4.3. Interaction by Child Age, Sex, Ethnic Background and SES

We also explored the associations between the number of SLEs and emotional, and behavioral problems by child age, gender, ethnic background, and family SES. Contrary to our hypotheses, these sociodemographic characteristics didn’t moderate the association between the number of SLEs and problem behaviors in this study. However, due to the small numbers of these subgroups, our results have to be interpreted with care. An earlier review by Granter provided inconclusive evidence on the moderating effects of these sociodemographic characteristics in the association between a stressor and psychopathology [12]. Stress is the product of the interplay of three components: the stressor, appraisal (i.e., perception), and the response to the stressor [37]. Therefore, not all children exposed to SLEs would suffer from the adverse outcomes caused by SLEs. For instance, children from disadvantaged families may be exposed to more stressors, and may have limited resources and coping skills to deal with these stressors, and would be more vulnerable to the SLEs [18,38]. However, the adverse effect of SLEs could be buffered if a supportive environment (e.g., social support, peer environment, and positive events) was provided [12]. Similarly, although children with a migrant background could perceive unique cultural stress brought by the changing context; specific cultural factors could also buffer the stress. However, our study did not evaluate potential mediators or the effect of positive factors. In line with previous studies [12,21], we recommend future studies using mixed methods (e.g., both qualitative and quantitative studies) to examine the pathways that underline the association between SLEs and child emotional and behavioral health; taking into the context in which events take place and are coped with.

### 4.4. Methodological Considerations

Strengths of this study include the relatively large sample size, the use of a validated questionnaire to assess emotional and behavioral problems, and the community-based setting.

However, this research has limitations. First, the causality of the associations cannot be asserted due to the cross-sectional design of the study. Second, SLEs were not measured by the validated tools. In addition, due to the relatively young age of the children, parental reports were used. Parents rated the presence and severity of each stressful life event, the impact on the child might have been different, i.e., more or less stressful. As a result, the association between impact and CBCL might be underestimated. Third, we used CBCL 1.5–5 to assess emotional and behavioral problems in our sample to enhance comparability. However, about 25.1% of the children fell outside this age boundary (21.5% were younger than 1.5, and 3.6% were older than six years old). In our sample, for the total scale, the externalizing and internalizing broadband scales, Cronbach’s alphas were comparable in 5-year-old children (0.94, 0.90, and 0.85) and children younger (0.93, 0.89, and 0.89) or older (0.95, 0.91, and 0.89) than five years, indicating that problems might also be reliably measured in children younger and older than five years. Fourth, although children’s scores on the CBCL scale were rather comparable with other studies [39,40], the non-respondent families were those with lower socioeconomic backgrounds (i.e., less educated, unemployed, lower income), living alone, and with a non-Dutch ethnic background. Previous studies have suggested that families with these characteristics are more vulnerable to the adverse effects of SLEs [12,18]. We recommend that future studies include a diverse sociodemographic population of parents and children. Finally, we used ‘number of SLEs’ to assess the burden of SLEs. However, as mentioned above, this approach assumes that risk was designated equally across events. We addressed this limitation by using a severity-weighted score in our sensitivity analysis. Nevertheless, other characteristics, such as the types, chronicity, and frequency of the events may also lead to differential impacts on the emotional and behavioral problems of children [21]. Future research exploring the SLEs and health outcomes would benefit from involving information on the types, duration, frequency of SLEs, and factors that may protect the child from the negative impact of these events.

## 5. Conclusions

The results of this study showed that parent-reported SLEs in the past year were associated with increased CBCL total, externalizing, and internalizing problem scale scores in a community sample of children 0–7 years old. The study supported the hypothesis that the impact of SLEs varies by event. Exposure to “work-family spillover”, “conflicts with neighbors, friends, acquaintances, or family”, “problems in the marriage”, and “divorce” were associated with relatively high levels of emotional and behavioral problems. We recommend longitudinal studies that are able to consider the timing, frequency, duration, severity, and co-occurrence of SLEs to confirm our findings, and to elucidate the possible mechanisms underlying these associations. Our findings underline the importance for health professionals to be aware of the impact of stressful life events on child emotional and behavioral well-being. While trying to reduce these events, when applicable, interventions supporting children and parents to manage and cope with SLEs should also be offered to minimize the impact.

## Figures and Tables

**Table 1 ijerph-19-01650-t001:** Descriptive statistics for demographics and problem behaviors (N = 959).

Characteristics	Missing (%)	All Children *n* = 959	No SLEs(0) *n* = 232	Low SLE(1) *n* = 246	Medium SLEs(2) *n* = 221	High SLEs(≥3) *n* = 260	*p* Value
**Demographic characteristics**							
Respondents (%)	0.0						0.538
Mother		863 (90.0)	211 (90.9)	226 (91.9)	194 (87.8)	232 (89.2)	
father		67 (7.0)	13 (5.6)	16 (6.5)	17 (7.7)	21 (8.1)	
Mother and father		29 (3.0)	8 (3.4)	4 (1.6)	10 (4.5)	7 (2.7)	
Respondents’ age (mean (SD)) (range 20–55)	0.0	34.1 (5.1)	34.4 (4.8)	33.9 (4.8)	34.0 (5.5)	34.0 (5.4)	0.737
Respondents’ educational level (%)	0.1						0.527
Low		64 (6.7)	21 (9.1)	13 (5.3)	16 (7.2)	14 (5.4)	
Medium		353 (36.8)	81 (34.9)	86 (35.1)	87 (39.4)	99 (38.1)	
High		541 (56.5)	130 (56.0)	146 (59.6)	118 (53.4)	147 (56.5)	
Family composition (one parent, %)	0.4	54 (5.7)	11 (4.8)	12 (4.9)	10 (4.6)	21 (8.1)	0.258
Household income (%)	6.4						0.404
Low (<3200)		287 (32.0)	64 (30.0)	67 (29.1)	67 (31.9)	89 (36.3)	
Medium (3200–4200)		368 (41.0)	98 (46.0)	94 (40.9)	86 (41.0)	90 (36.7)	
High (>4200)		243 (27.1)	51 (23.9)	69 (30.0)	57 (27.1)	66 (26.9)	
Child age (years/mean (SD)) (range 0–7)	0.9	3.3 (1.9)	3.2 (1.8)	3.3 (1.9)	3.4 (1.9)	3.2 (2.0)	0.528
Child ethnic background (non-Dutch, %)	1.7	112 (11.9)	22 (9.6)	27 (11.1)	24 (11.1)	39 (15.4)	0.215
Child gender (girls, %)	0.4	456 (48%)	109 (47.0)	115 (46.7)	109 (49.5)	123 (47.9)	0.932
**Emotional- and behavioral** **problems ^#^**							
CBCL-T (mean (SD))	0	20.3 (16.7)	16.3 (15.0)	19.5 (15.6)	20.9 (17.2)	24.2 (18.0)	<0.001
Borderline problem (>46) (%) ^#^		77 (8.0)	13 (5.6)	12 (4.9)	20 (9.0)	32 (12.3)	0.008
Clinical problem (CBCL-T>56) (%)		38 (4.0)	4 (1.7)	5 (2.0)	9 (4.1)	20 (7.7)	0.002
CBCL-I (mean (SD))	0	4.8 (5.3)	3.8 (4.7)	4.4 (4.7)	5.3 (5.8)	5.7 (5.6)	<0.001
Borderline problem (>12) (%)		73 (7.6)	14 (6.0)	11 (4.5)	18 (8.1)	30 (11.5)	0.018
Clinical problem (CBCL-I > 18) (%)		43 (4.5)	7 (3.0)	6 (2.4)	12 (5.4)	18 (6.9)	0.054
CBCL-E (mean (SD))	0.1	8.8 (7.1)	7.2 (6.4)	8.8 (7.1)	8.8 (7.2)	10.4 (7.4)	<0.001
Borderline problem (>18) (%)		102 (10.6)	14 (6.1)	30 (12.2)	25 (11.3)	33 (12.7)	0.073
Clinical problem (CBCL-E > 23) (%)		42 (4.4)	5 (2.2)	9 (3.7)	10 (4.5)	18 (6.9)	0.071

^#^ The cutoff points were based on the 83rd percentile of a Dutch norm group. CBCL-T = Child Behavior Checklist total scale score, CBCL-E = Child Behavior Checklist externalizing scale score, CBCL-I = Child Behavior Checklist internalizing scale score; SLE= stressful life events; the continuous variables were tested by one-way Anova, and the categorical variables were tested by the Chi-square test.

**Table 2 ijerph-19-01650-t002:** Frequency and severity rating of the life events among the participants (N = 959).

Stressful Life Events	Occurrence		If Yes, Severity Rating		
	No (*n*, %)	Yes (*n*, %)	Not at All Stressful	Somewhat Stressful	Highly Stressful
(1) Moving to another address	774 (80.7)	185 (19.3)	83 (44.9)	77 (41.6)	25 (13.5)
(2) A friend of the child moving to another address	914 (95.3)	45 (4.7)	35 (77.8)	6 (13.3)	4 (8.9)
(3) Tension at the parents’ work that has been felt at home	606 (63.2)	353 (36.8)	100 (28.3)	216 (61.2)	37 (10.5)
(4) Financial problems	875 (91.2)	84 (8.8)	25 (29.8)	46 (54.8)	13 (15.4)
(5) Conflicts with neighbors, friends, acquaintances, or family	850 (88.6)	109 (11.4)	40 (36.7)	56 (51.4)	13 (11.9)
(6) Fire or burglary	935 (97.5)	24 (2.5)	18 (75.0)	6 (25.0)	-
(7) Problems with the physical health of people in close proximity	612 (63.8)	347 (36.2)	139 (40.0)	163 (47.0)	45 (13.0)
(8) Problems with the psychological health of people in close proximity	770 (80.3)	189 (19.7)	71 (37.6)	89 (47.1)	29 (15.3)
(9) Death of someone in close proximity	745 (77.7)	214 (22.3)	109 (50.9)	73 (34.1)	32 (15.0)
(10) Problems in the marriage relations	884 (92.2)	75 (7.8)	19 (25.3)	38 (50.7)	18 (24.0)
(11) Divorce	933 (97.3)	26 (2.7)	8 (30.8)	11 (42.3)	7 (26.9)
(12) Unemployment	886 (92.4)	73 (7.6)	43 (58.9)	27 (37)	3 (4.1)

**Table 3 ijerph-19-01650-t003:** Association between the exposure to single stressful life events (no/yes) and emotional and behavioral problems in young children age 0–7 years old (N = 959).

Individual Stressful Life Events	CBCL-T β (95% CI)		CBCL-I β (95% CI)		CBCL-E β (95% CI)	
	Model 1	Model 2	Model 1	Model 2	Model 1	Model 2
(1) Moving to another address	0.25 (−2.53, 3.04)	−1.16 (−3.93, 1.62)	−0.02 (−0.86, 0.82)	−0.39 (−1.24, 0.45)	−0.15 (−1.37, 1.06)	−0.73 (−1.94, 0.47)
(2) A friend of the child moving to another address	0.43 (−4.80, 5.66)	−3.09 (−8.36, 2.19)	−0.08 (−1.66, 1.50)	−1.01 (−2.61, 0.59)	0.34 (−1.93, 2.61)	−1.05 (−3.35, 1.25)
(3) Tension at the parents’ work that has been felt at home	** *5.62 (3.39, 7.84)* **	** *4.52 (2.22, 6.83)* **	** *1.35 (0.67, 2.02)* **	** *1.04 (0.34, 1.75)* **	** *2.45 (1.48, 3.42)* **	** *2.05 (1.05, 3.06)* **
(4) Financial problems	** *6.12 (2.05, 10.19)* **	3.40 (−0.78, 7.58)	** *1.94 (0.72, 3.17)* **	** *1.29 (0.02, 2.56)* **	** *2.04 (0.27, 3.81)* **	0.91 (−0.92, 2.73)
(5) Conflicts with neighbors, friends, acquaintances or family	** *6.22 (2.69, 9.75)* **	** *4.61 (1.05, 8.18)* **	** *1.82 (0.75, 2.89)* **	** *1.42 (0.34, 2.50)* **	** *2.35 (0.81, 3.89)* **	** *1.70 (0.14, 3.25)* **
(6) Fire or burglary	2.83 (−4.73, 10.40)	−1.19 (−8.77, 6.39)	1.06 (−1.22, 3.35)	0.04 (−2.26, 2.34)	1.51 (−1.77, 4.80)	−0.07 (−3.37, 3.23)
(7) Problems with the physical health of people in close proximity	1.64 (−0.61, 3.88)	−0.15 (−2.44, 2.15)	0.36 (−0.32, 1.04)	−0.11 (−0.81, 0.59)	0.73 (−0.24, 1.71)	0.04 (−0.96, 1.04)
(8) Problems with the psychological health of people in close proximity	** *3.22 (0.54, 5.91)* **	1.32 (−1.43, 4.08)	0.75 (−0.06, 1.57)	0.25 (−0.59, 1.08)	** *1.54 (0.38, 2.71)* **	0.83 (−0.37, 2.03)
(9) Death of someone in close proximity	** *3.63 (1.03, 6.23)* **	2.42 (−0.19, 5.04)	** *0.85 (0.07, 1.64)* **	0.53 (−0.26, 1.32)	** *1.43 (0.3, 2.56)* **	0.95 (−0.19, 2.09)
(10) Rroblems in the marriage relations	** *6.49 (2.49, 10.49)* **	4.08 (−0.01, 8.17)	** *1.92 (0.71, 3.13)* **	** *1.33 (0.08, 2.57)* **	** *2.59 (0.85, 4.33)* **	1.63 (−0.16, 3.41)
(11) Divorce	** *12.77 (5.02, 20.52)* **	** *8.23 (0.33, 16.12)* **	** *5.63 (3.30, 7.95)* **	** *4.65 (2.26, 7.03)* **	2.83 (−0.55, 6.22)	0.83 (−2.62, 4.27)
(12) Unemployment	2.62 (−1.60, 6.84)	0.12 (−4.14, 4.38)	0.31 (−0.97, 1.59)	−0.37 (−0.66, 0.92)	1.06 (−0.77, 2.90)	0.07 (−1.78, 1.93)

Notes: Model 1: adjusted for child age, sex, ethnic-background, respondents’ age, sex, educational level, family composition, household income, and recruitment methods; Model 2: additionally, adjusted for the total number of other SLEs; CBCL-T = Child Behavior Checklist total scale score, CBCL-E = Child Behavior Checklist externalizing scale score, CBCL-I = Child Behavior Checklist internalizing scale score; bold and italic indicates *p* < 0.05.

**Table 4 ijerph-19-01650-t004:** Association between multiple stressful life events (number of SLEs) and emotional and behavioral problems in children aged 0–7 years old (*n* = 959).

	CBCL-T		CBCL-E		CBCL-I	
Number of SLEs ^*^	β	Bootstrapped 95% CI	β	Bootstrapped 95% CI	β	Bootstrapped 95% CI
0 (*n* = 232)	1.00 (ref.)	1.00 (ref.)	1.00 (ref.)	1.00 (ref.)	1.00 (ref.)	1.00 (ref.)
1 (*n* = 246)	** *3.86* **	** *(0.96, 6.70)* **	** *1.95* **	** *(0.67, 3.20)* **	0.73	(−0.11, 1.57)
2 (*n* = 221)	** *4.29* **	** *(1.45, 7.29)* **	** *1.57* **	** *(0.33, 2.90)* **	** *1.30* **	** *(0.43, 2.26)* **
≥3 (*n* = 260)	** *7.10* **	** *(4.36, 10.14)* **	** *3.12* **	** *(1.86, 4.50)* **	** *1.59* **	** *(0.75, 2.46)* **

Note: * Number of SLEs was calculated by counting all SLEs that an individual had experienced in the past 12 months; CBCL-T = Child Behavior Checklist total scale score, CBCL-E = Child Behavior Checklist externalizing scale score, CBCL-T = Child Behavior Checklist internalizing scale score; adjusted for child age, sex, ethnic-background, respondents’ age, sex, educational level, family composition, household income, and recruitment methods; SLE = stressful life event; bold and italic indicates *p* < 0.05.

## Data Availability

The data that support the findings of this study are available from the corresponding author (H.R.), upon reasonable request.

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
