# Peer review of "The Association between Stressful Life Events and Emotional and Behavioral Problems in Children 0–7 Years Old: The CIKEO Study"

_ijerph, 2022, doi:10.3390/ijerph19031650_

Round 1
Reviewer 1 Report
This study attempts to establish the relationship between SLE and the risk of emotional and behavioural problems in children between 0 and 7 years of age based on parental information and through a cross-sectional design.
The interest of this study lays in the useful data it provides regarding the improvement of the understanding of how early experiences affect the psychological development and mental health of the population. At the same time, it contributes to improving intervention programs by identifying stressful contexts that may be critical to their later mental health.
As the authors themselves point out, there are several aspects that limit the generalization of their results, and that could be decisive to understand when a stressful event is at the base of the appearance of an emotional or behavioural problem, or when, despite it, no negative effects are produced on the individual.
As this is an investigation whose source of data is parental reports, in subsequent studies it might be necessary to consider that some characteristics of the parents may influence those reports. Some of them may be parental beliefs and expectations about the development and education of their children, or if they are families with children or other dependent relatives or with developmental disorders.
On the other hand, stressful events do not always give rise to the appearance of psychopathological problems. We cannot know if there are cases of resilient children within the sample (for example, within the migrant population). In these cases, the satisfaction of Basic Psychological Needs, from the perspective of self-determination theory, for example, could explain this type of resilience in the face of adverse situations. More parental information about the satisfaction of basic psychological needs could be useful for future research.
On the other hand, the study has strengths that, in my opinion, justify its publication, and its usefulness must be based on the quality of the data provided and the adequacy of the analyses carried out.
Thus, for example, the theoretical framework clearly establishes the research background through relevant primary sources, as well as the conceptual limits of the research. This is especially important given the wide range of variables that play a role in modulating the effects that stressful events have on emotions and behaviour.
From the methodological point of view, it uses an appropriate design and adequately defines the sample, the variables and instruments used in the study.Also, the analysis of the data is appropriate and the results are relevant to the research objectives.
The discussion and conclusions fit the data and connect appropriately with the preceding research.
The authors clearly identify the limitations of the study and the future direction of the research.
Therefore, I consider that this study can be accepted for publication without modifications.
Author Response
We sincerely thank the reviewer for the positive feedback on the manuscript.
Reviewer 2 Report
The subject is interesting but the article has serious errors, lack of summary, inconsistencies in the development of the instruments and little discussion.
It uses non-validated tools as the authors themselves indicate in point 4.4. The instrument used is not adequately explained, point 2.4. (it indicates that it has 100 items and then that one subscale has 36 items and another has 24). Furthermore, the instrument used is for children from 1.5 to 5 years of age and in the title it indicates 0 to 7 years of age.
It talks about the 83rd percentile of a group of Dutch norms, but does not explain or indicate anything else, it would be appropriate to give an example.
The writing is very rambling and difficult to understand, in addition to not including the mandatory summary of any scientific article. During the reading it appears three times (see above), which section or point of the document it refers to.
Author Response
The subject is interesting but the article has serious errors, lack of summary, inconsistencies in the development of the instruments and little discussion.
- It uses non-validated tools as the authors themselves indicate in point 4.4.
Response of the authors: We agree with the reviewer the instrument we used for stressful life events is not valid. This is the limitation of the current study, which was mentioned at line 356 . The instrument used was based on existing literature on this topic [1].
- The instrument used is not adequately explained, point 2.4. (it indicates that it has 100 items and then that one subscale has 36 items and another has 24).
Response of the authors: In this study, we focused on the total CBCL score and the scores on the Internalizing and externalizing broadband scales. To clarify, we adapted the text.
Child emotional and behavioral problems Emotional and behavioral problems were assessed using the 100-item Child Behavior Checklist for ages 1.5 to 5 (CBCL) [26, 27]. As the majority (73.9 %) of the children included in the current study fell in the range of 1.5-6 years old at the time of CBCL assessment (21.5 % were younger than 1.5 and 3.6 % of the children were older than six years old), we used the CBCL 1.5–5 version for all children to enhance comparability [28].
Parents rated the occurrence of their child’s behavior within the past 2 months on a three-point scale with 0 (not true) and 2 (very true or often true). The CBCL includes a total problem score taking all 100 items (CBCL-T). two broadband scales can be calculated with a selection of items: an internalizing and externalizing scale. The Internalizing scale (CBCL–I) is calculated from the 36 items that-assess behaviors such as withdrawal, anxiety, and depression; the Externalizing scale (CBCL–E) is calculated by the 24 items -assessing behaviors, such as attention problems and aggressive behavior. A weighted sum score allowing for 20% of missing data was calculated for the total scale and the two broadband scales by summing the items belong to each (sub)scale. Higher scores indicate higher levels of problems. To illustrate the clinical significance, we also calculated the percentage of children with a score above the borderline and clinical range. The cut-off points were based on the 83rd percentile based on a Dutch norm group (Table 1) [29].
- Furthermore, the instrument used is for children from 1.5 to 5 years of age and in the title it indicates 0 to 7 years of age.
Response of the authors: We thank the reviewer for pointing this out. This issue has been discussed in section 2.4 Measures and 4.1 Methodological Considerations.
In section 2.4 Measures : “As the majority (73.9 %) of the children included in the current study fell in the range of 1.5-6 years old at the time of CBCL assessment (21.5 % were younger than 1.5 and 3.6 % of the children were older than six years old), we used the CBCL 1.5–5 version for all children to enhance comparability [2].”
In section 4.1 Methodological Consideration: “Third, we used CBCL 1.5-5 to assess emotional and behavioral problems in our sample to enhance comparability. However, about 25.1% of the children fell outside this age boundary (21.5% were younger than 1.5, and 3.6% were older than six years old). In our sample, for the total scale, the externalizing and internalizing subscales, Cronbach’s alphas were comparable in 5-year-old children (0.94, 0.90, and 0.85) and children younger (0.93,0.89, and 0.89) or older (0.95,0.91 and 0.89) than five years, indicating that problems might also be reliably measured in children younger and older than five years.”
- It talks about the 83rd percentile of a group of Dutch norms, but does not explain or indicate anything else, it would be appropriate to give an example.
Response of the authors: We thank the reviewer’s suggestion. Details regarding the cut-off points used was given in Table 1. We added a sentence to the methods section to refer to the details in Table 1.
“The cut-off points were based on the 83rd percentile based on a Dutch norm group (Table 1)”
- The writing is very rambling and difficult to understand, in addition to not including the mandatory summary of any scientific article.
Response of the authors: We apologize for the missing abstract, it has been added (please see below for more details). We have done a thorough editorial read of the manuscript and adjusted typographical errors and improved readability.
Abstract
Background: Stressful life events (SLEs) are recognized risk factors for emotional and behavioral problems, but the association is understudied among young children. Our aim was to examine the association between exposure to SLEs and emotional and behavioral problems in young children up to 7 years old. Methods: We analyzed baseline data from 959 children (mean age = 3.3 years; SD = 1.9; 47.5% girls) in the CIKEO study, a community-based longitudinal study in the Netherlands. Linear regression was used to assess the associations between the total as well as the individual exposure to SLEs experienced in the past 12 months and emotional and behavioral problems assessed by CBCL 1.5-5. Interaction of SLEs and child age, sex, ethnic background, and socio-economic status was explored. Results: Higher total exposure to SLEs, as indicated by the number of SLEs, was significantly associated with higher CBCL total, internalizing and externalizing problem scores (p for trend <0.05). The results did not differ by child age, sex, ethnic background or family SES. Six out of the 12 SLEs explored were independently associated with greater CBCL total/externalizing/internalizing scores (p<0.05). Conclusions: Exposure to SLEs is associated with higher levels of emotional and behavioral problems in young children. And the impact of SLEs may vary by events. Stressful life events might be a target for interventions to improve emotional and behavioral well-being among young children.
- During the reading it appears three times (see above), which section or point of the document it refers to.
Response of the authors: We apologize for the confusion caused. After discussion, we decide to remove the ‘see above’ in the text.
Reviewer 3 Report
Thank you for facilitating the review.
The article is original and relevant to know the level of stress and its repercussions in young children.
All parts of the article are well designed and the information is relevant.
Its publication is recommended when the abstract is written and the design of the study is indicated with greater precision.
Also, check the references. In several cases the first author et al. when the number of authors does not justify this format (for example ref,1)
Author Response
Thank you for facilitating the review.
The article is original and relevant to know the level of stress and its repercussions in young children. All parts of the article are well designed and the information is relevant.
- Its publication is recommended when the abstract is written and the design of the study is indicated with greater precision.
Response of the authors: we apologize for the missing abstract; it has been added to the manuscript. (Please see below).
Abstract
Background: Stressful life events (SLEs) are recognized risk factors for emotional and behavioral problems, but the association is understudied among young children. Our aim was to examine the association between exposure to SLEs and emotional and behavioral problems in young children up to 7 years old. Methods: We analyzed baseline data from 959 children (mean age = 3.3 years; SD = 1.9; 47.5% girls) in the CIKEO study, a community-based longitudinal study in the Netherlands. Linear regression was used to assess the associations between the total as well as the individual exposure to SLEs experienced in the past 12 months and emotional and behavioral problems assessed by CBCL 1.5-5. Interaction of SLEs and child age, sex, ethnic background, and socio-economic status was explored. Results: Higher total exposure to SLEs, as indicated by the number of SLEs, was significantly associated with higher CBCL total, internalizing and externalizing problem scores (p for trend <0.05). The results did not differ by child age, sex, ethnic background or family SES. Six out of the 12 SLEs explored were independently associated with greater CBCL total/externalizing/internalizing scores (p<0.05). Conclusions: Exposure to SLEs is associated with higher levels of emotional and behavioral problems in young children. And the impact of SLEs may vary by events. Stressful life events might be a target for interventions to improve emotional and behavioral well-being among young children.
For the current study, we used baseline data of parents and children participating the CIKEO study. This was indicated in the abstract and study design.
In the abstract: “We analyzed baseline data from 959 children (mean age = 3.3 years; SD = 1.9; 47.5% girls) in the CIKEO study, a community-based longitudinal study in the Netherlands.”
In 2.1 study design and data collection: “In this study, we applied a cross-sectional design using the baseline data from the CIKEO study.” (line 103-104).
- Also, check the references. In several cases the first author et al. when the number of authors does not justify this format (for example ref,1)
Response of the authors: We thank the reviewer for noticing these citation issues and have changed them to the reference style for MDPI ACS Journals.
Round 2
Reviewer 2 Report
Previous errors have been corrected.
Author Response
Thank you very much